# Food-Based Dietary Guidelines for Infants in Latin America and the Caribbean: A Systematic Review

**DOI:** 10.3390/nu16081233

**Published:** 2024-04-21

**Authors:** Isabelle Cristina Daniel, Mariana Sofia Moro Siqueira, Gabriela Ulbricht Romaneli, Juliana Schaia Rocha Orsi, Renata Iani Werneck

**Affiliations:** Graduate Program in Dentistry, School of Medicine and Life Sciences, Pontifícia Universidade Católica do Paraná, Curitiba 80242 980, Brazil; cristina.isabelle@pucpr.edu.br (I.C.D.); gabriela.romaneli@pucpr.edu.br (G.U.R.); juliana.orsi@pucpr.br (J.S.R.O.)

**Keywords:** food-based dietary guidelines, infant nutrition, food sustainability, national dietary recommendations, health promotion, public health

## Abstract

Food-based dietary guidelines (FBDGs) are tools for promoting healthy eating habits. For the population of children under two years old in Latin America and the Caribbean (LAC), there is a lack of reviews analyzing the quality of these guidelines. The objective of this systematic review is to evaluate publicly available FBDGs for the population under two years old in LAC until mid-2023. Guidelines aimed at caregivers of children were included, sourced from government websites in LAC countries and the Food and Agriculture Organization (FAO) portal. Documents targeted at healthcare professionals were excluded. For qualitative analysis, the Agree II guidelines assessment tool and the FAO guide principles for developing healthy and sustainable diets were used. The results showed that more recently released and revised FBDGs with a greater number of pages obtained better scores in both assessments. Additionally, out of the 32 LAC countries, only 13 had these FBDGs available on websites for public access. As a limitation, this study faced challenges in standardizing the searches on government websites. The authors emphasize the need to develop FBDGs for the population under two years old that align with current health and sustainability needs and promote health education.

## 1. Introduction

Food and nutrition policies are promoted worldwide by various public health sectors as a means of reducing the incidence of preventable diseases. One of the strategies employed is nutrition education, which involves using tools like food-based dietary guidelines (FBDGs). These guidelines should feature illustrative schemes that are easy to understand, helping individuals to make necessary adjustments to their dietary habits. However, many hurdles make the effective implementation of these guides a challenge [1], and reviews of the literature have shown that various populations around the world have an insufficient degree of comprehension and use of the food guide contents to the extent of changing their behaviors [2,3]. Strategies are required to overcome this challenge by being assertive in the methods of developing FBDGs and thereby transmitting recommendations.

The main potential of these documents resides in changing dietary habits towards a more healthy, sustainable, and respectful collective eating style for the various global food cultures [4]. However, currently, there is a gap in terms of food-based dietary guidelines’ advice on breastfeeding and complementary feeding that is appropriate for the infant population [5,6], though the first two years of life present a window of opportunity in the establishment of healthy eating behaviors [7].

In public health, the delay of child development—given the specific needs of this age group—indicates socioeconomic vulnerability, as well as the sensitivity of child development to external social determinants, including poor nutrition [7]. For Latin America and the Caribbean (LAC), the Food and Agriculture Organization of the United Nations (FAO), in its 2022 report, presented a growing profile of childhood obesity among children under 5 years old, of which there is a percentage higher than the global average, with South America having the highest rates [8]. In this report, it was also shown that 22% of this collective population cannot afford a healthy diet due to the cost. This inaccessibility is directly related to the poverty index, and it has the greatest impact on vulnerable populations, including children [9,10]. Henceforth, it is justified to prioritize public policies aimed at the maternal and child populations, as well as studies and research focused on their health needs [5,7]. 

Several studies have reviewed the status of the development and implementation of FBDGs by various countries in recent years [1,6,11,12]. Nevertheless, there is a notable absence of studies on the specific population of children under two years old, particularly in Latin America and the Caribbean [1,5,6,13]. This research aims to review the FBDGs directed toward the population under two years old that are available in Latin America and the Caribbean and to evaluate how their guidelines align with current health, sustainability, and social participation needs. 

## 2. Materials and Methods

This systematic review followed the PRISMA [14] (Preferred Reporting Items for Systematic Review and Meta-Analyses) protocol and was registered on PROSPERO: CRD42023421883. The research question was “What are the existing food-based dietary guidelines for children under two years old in Latin America and the Caribbean that support food and nutrition education for the resident population?” To structure the question, the PiCo acronym was used, as per the recommendation for reviews of policies [15], consisting of the following parts:Population (P): the caregivers of infants;Interest (I): food-based dietary guidelines for children under two years old;Context (Co): the countries of Latin America and the Caribbean.

### 2.1. Search Strategy and Selection

Two reviewers (I.C.D. and M.S.M.S.) searched independently in the gray literature. The kappa Cohen [16] for statistical calculations was used to assess agreement between the reviewers.

The search for documents was conducted through online databases of the FAO (Food and Agriculture Organization of the United Nations), PAHO/WHO (Pan American Health Organization), and LILACS (Latin America and the Caribbean Literature on Health Science), filtering the countries of Latin America and the Caribbean [17], until the midpoint of 2023 (29/06). To ensure the accuracy of the databases used, the official government portals of the countries (Appendix A) without guidelines included in the mentioned databases were also checked. Additionally, these websites were searched for all countries to identify updates to the guidelines. If FBDGs for infants in a country were not found on the FAO, PAHO, or LILACS databases or government portals, it was concluded that no document was available; therefore, the country was not included in this review.

The search terms used in the portals were “food based dietary guidelines for children under two years old” and the equivalent terms in Spanish and Portuguese (Guías Alimentarias para ninõs y niñas menores de dos años/Guias Alimentares para crianças menores de dois anos).

### 2.2. Eligibility Criteria: Inclusion and Exclusion

Food guides were included without any limitation on language, so long as they were launched by the first half of 2023. The inclusion criteria were official documents from the Departments of Health, Food, and Social Services, and similar entities, of the 32 countries that comprise LAC. These documents had titles such as “Food Guide for Infants and Young Children under 2 Years of Age”, “The Pregnancy Food Guide”, “Food Guide for Babies”, or “The Family Food Guide”. We excluded FBDGs that were published for health workers, non-government organizations or institutions, or any other stakeholders besides the caregivers of infants.

### 2.3. Data Extraction

Two reviewers (I.C.D. and M.S.M.S.) collected the food guides independently as recommended by the PRISMA protocol. The screening process involved selecting documents that referred directly or indirectly to the population of children under two years old. Afterward, for analyses, the documents were read in full following the exclusion criteria. To read the documents in full, the tool Google Translator was used to translate the guides from Spanish to Portuguese.

The data collected from the FBDGs for the first analysis consisted of the criteria composing the tool Agree II [18], specified in “scope and purpose”, “stakeholder involvement”, “rigor of development”, “clarity of presentation”, “applicability” and “editorial independence”. For the second analysis, the key recommendations of the FBDGs were studied in terms of health, environmental impact, and sociocultural factors according to the Sustainable Healthy Diets Guiding Principles [19].

### 2.4. Data Analysis 1: Agree II

To assess the quality of food-based dietary guidelines (FBDGs), three reviewers (nutritionists) conducted an independent analysis using the Agree II tool (Appraisal of Guidelines for Research & Evaluation II). This tool evaluates the methodological rigor and transparency of the development of these guidelines by examining the documents from the perspective of policy developers. The reviewers’ IDs are I.C.D., M.S.M.S., and G.U.R. The tool consists of assessment domains, rated on a 7-point scale per item, and the total is scaled as a percentage. The overall assessment was judged based on an original scale constructed (Appendix A) that supports objectively building the overall perception of the guides and the general point of view of the reviewers on the presentation and cohesiveness of advice in the FBDGs. This scale was constructed by categorizing the maximum score (600%) across six domains into seven mirror Likert scale categories. The overall assessment was generated by ranking the average of the reviewers’ ratings.

For the analysis of documents using the Agree II tool (Portuguese version), Excel software (version 2013) was used to calculate scores for comparisons between the guidelines according to the formula outlined in the Agree II protocol. These analyses were presented in tabular format.

To assess agreement among reviewers during the Agree II analysis, the Fleiss kappa test was utilized, given that three assessors were involved, using SPSS software version 29.0.1.0 (171). The Likert scale for Agree II scoring ranges from one to seven, with ambiguous ratings of “partially agree” and “partially disagree”. To prevent accidental disagreements, two calibration assessments were conducted among the reviewers. The presence of a fourth reviewer was considered in case of disagreement among the three reviewers’ assessments.

### 2.5. Data Analysis 2: Sustainable Healthy Diets Guiding Principles

The discussions were supported by qualitative analyses of key recommendations in the FBDGs, presented in a frame format. In this appraisal, the document “Sustainable Healthy Diets Guiding Principles” [19] elaborated by the FAO was used, following the aspects of health, environmental impact, and sociocultural factors. The key advice of each guide was evaluated by two reviewers (I.C.D. and M.S.M.S.) according to agreement with the Sustainable Healthy Diets Guiding Principles, signaled in a signal pattern. In this way, the guides whose items matched the aims aspects by explaining the concept and providing orientation in their recommendations were matched in green. If the key recommendation only explained or only oriented the aspect, it was marked in yellow; if the recommendation did not, it was marked in red. The assessment was conducted collaboratively, and the final result represented the consensus reached by the reviewers. The sought aspects were categorized as stated in the document into the sets “Health Aspects”, “Environment Impact Aspects”, and “Sociocultural Aspects”.

## 3. Results

Sixteen FBDGs for the population under 2 years old from the 32 countries of Latin America and the Caribbean were selected; six were found on the FAO portal [20], and nine were located on the official government websites of the countries (references and links of portals accessed are available in Appendix A). Two reviewers selected the documents, and the Kappa test showed near-perfect agreement [16] among the observers (k = 0.934; *p* < 0.001). After the screening of the introduction and objectives, three documents were excluded because they were intended for healthcare teams. In the end, 13 dietary guidelines were analyzed using Agree II. One document had initially been excluded for being intended for healthcare professionals, but on the website, it was possible to find the version for the population of interest in this dietary guide [21]. Therefore, the guide was manually added. The flowchart of the document adapted to gray literature selection is presented in Figure 1.

Among the selected guidelines, 11 are from South American countries, 1 is from Central America, and 1 is from the Caribbean, with document presentation dates ranging from 2003 to 2020. Regarding content, the smallest identified guide had 22 pages [22], while the largest had 265 [23]. The main dietary recommendations varied from 7 [24] to 19 key items [25]. The distribution of the identified guidelines is presented in Table 1.

In general, all FBDGs presented traditional recommendations regarding the nutritional aspects of eating patterns. Some documents included advice related to dietary culture [35] in their key recommendations, which can be highlighted as follows:Argentina: “Love: an important and necessary food”;Bolivia: “Traditional ancestral foods”;Brazil: “Ensure that mealtime is a moment of positive experiences, learning, and affection with the family” and “Protect the child from advertising targeting children”;Chile: “In the breastfeeding process, the father’s participation is essential, supporting the mother, strengthening attachment, assisting with other household tasks, or caring for other children, and holding the child between feedings”;Colombia: “As a lactating woman, you have the right to have your partner, family, and society support you to make breastfeeding a successful practice.”

### 3.1. Assessment through Agree II

For the analysis of the documents with the Agree II tool, the Excel software feature was used to compute scores for comparison between the guidelines following the formula outlined in the Agree II protocol. The Fleiss kappa was calculated to assess the agreement between the reviewers, with a coefficient of 0.470 indicating moderate agreement [16] (95% CI; 0.268–0.672; z = 4.551; *p* < 0.001). The qualifications used are shown in Table 2.

The scores were calculated as the mean of all reviewers’ evaluations, resulting in an adequacy percentage representing how well each domain was addressed in each guide according to the reviewers’ perspectives. The scores could range from 0% to 100% adequacy.

#### Overall Guideline Assessment

For the overall evaluation, the performance of each dietary guideline in the analysis of the aforementioned domains was considered with adequacy categorized from one (lowest possible quality) to seven (highest possible quality). The average rating obtained among the documents was five, while that obtained for the observed mode was four. The highest score was found in the dietary guidelines from Brazil (seven), followed by the guidelines from Colombia (six), Peru (six), and Uruguay (six). The lowest category obtained was four, which was observed in the documents from Bolivia, Guatemala, Mexico, the Dominican Republic, and Venezuela.

### 3.2. Assessment through the Sustainable Healthy Diets Guiding Principles

The main recommendations of each FBDG were analyzed according to their alignment with the objectives of Healthy and Sustainable Diets Guidelines (FAO, 2019). An illustration of these assessments is provided in Table 3.

The only aspect highlighted in green in all food guides was the adequate consumption of energy (calories) and nutrients. The generally well-highlighted items included recommendations for exclusive breastfeeding for up to 6 months, consumption of fruits and vegetables, prevention of noncommunicable chronic diseases (NCDs), respect for local dietary culture, and proper hygiene.

In contrast, the aspect of consuming foods with antibiotics and hormones was not mentioned by any country, and the item promoting gender-equitable participation was mentioned by only two countries (Colombia and Chile). Aspects with lower frequencies included recommendations to avoid foods with pesticides, strategies to manage food waste, guidance on reducing plastics in production and consumption, the moderate consumption of meats and eggs, and guidance on the consumption of processed and ultra-processed foods.

The country’s nomenclatures are abbreviated as follows: Argentina (ARG), Bolivia (BOL), Brazil (BRA), Chile (CHL), Colombia (COL), Guatemala (GUA), Mexico (MEX), Panama (PAN), Paraguay (PAR), Peru (PRU), the Dominican Republic (DOM), Uruguay (URU), and Venezuela (VEN).

## 4. Discussion

This review systematically analyzed food-based dietary guidelines for the population under 2 years old in Latin America and the Caribbean using the Agree II analysis tool and the Guiding Principles of Sustainable and Healthy Diets (FAO). The updated food guidelines with more content and pages scored higher in the Agree II evaluation, covering more health, social, and sustainability aspects in their recommendations. This information can be considered in new formulations and updates of FBDG processes to amply their potential for being connected with other sectors beyond health [4]. 

Of the 32 countries in Latin America and the Caribbean, less than half had available food guidelines with specific guidance for children under two years old. This tendency toward a less significant quantity has been previously noted in other systematic reviews and continues to be observed in this research [1,36,37]. Despite the increasing discussion on the importance of developing food guidelines, there is a notable need for greater encouragement in developing these documents to reach a larger population of infants and their caregivers [5,11].

Among the compiled guidelines, three had an abridged version or a short illustrated practical guide with specific guidelines for infant nutrition, and these documents had the highest number of pages. The quantitative breadth of a dietary guide can hinder access to reading, so producing short educational materials based on recommendations is an effective educational strategy [38,39]. However, there is no exact consensus in the literature regarding the use of only short images in the nutritional literacy of the population [40]. While guide images may be more accessible to a population with varying levels of education [41], they can also represent a simplification of the cultural diversity of food in a country [42]. Guides with more content offer greater potential for cross-sectoral implementation by providing more educational material in dialog with sectors beyond health [40,41]; therefore, this is the right strategy to be combined with abridged versions for educational actions. 

Regarding the Agree II evaluations, the least scored domain by the food guidelines was the methodological structure of the documents (domain three) and the reference in building the presented recommendations. Systematic reviews applying the tool observed similar results, with this domain showing some of the lowest adequacies compared to others [43,44,45,46]. These data show a weakness in presenting this scientific methodological support for the guidelines. Although the analyzed documents in this research are focused on caregivers of children under two years old, the information on methodological rigor is relevant for providing scientific support to the formulated recommendations, ensuring credibility to the food guidelines.

Through domains two (stakeholder involvement), four (clarity of presentation), and five (applicability) of the Agree II evaluation, the intelligibility of these food guidelines can be observed. Stakeholder involvement had an average adequacy of 56%, indicating participation by just over half, correlating with results from other Agree II reviews [43,44,45,46]. This domain includes an item related to the population’s understanding of the document, and a systematic review concluded that despite a growing trend in some populations’ knowledge about FBDGs, the degree is still low and does not automatically translate into understanding [12]. Thus, involving populations in the development of their food guides as a means to achieve a better comprehension of their content remains a challenge for most countries, as shown in this review.

Functional health literacy is the degree to which individuals can observe, process, and understand the information necessary for healthy lifestyle choices [39]. In this context, understanding the content of these food guidelines is crucial for guiding the health education of populations, aiming to build healthier eating habits [39]. Considering that the average evaluation of the clarity of presentation of the guidelines was 78% and the applicability was 56%, it is notable that the documents aim to provide accessibility to their content. However, few health literacy strategies are applied in the formulation of health educational materials [2], and in this Agree II evaluation, only the Food Guide of Brazil registered how the target population was reached.

The food guidelines well scored by the Agree II evaluation were from Brazil, Colombia, Peru, and Uruguay, and they also covered more health and sustainability aspects in their recommendations. In the evaluation of healthy and sustainable diet aspects, the most highlighted field covers health recommendations, where documents explain and provide guidance on healthy eating habits. The reviews of food guidelines that qualitatively assessed their recommendations also revealed that this guidance focused on food groups that consider human health but do not equally emphasize environmental sustainability [47,48].

The results revealed that the environmental impact aspect received the lowest evaluation among the three, especially the alert about consuming foods without excess hormones and antibiotics, which was not found in any of the food guidelines. Literature reviews seek to clarify the possible harms of consuming these substances used in the production of animal-origin foods, as well as their necessity in preserving human and animal health [49,50]. Despite their carcinogenic characteristics, the ingestion of hormone residues used for animal growth is a low risk to human health when used under responsible professional supervision due to the small quantities of hormones present in the products [49,50]. However, antibiotics pose a significant risk due to the potential for resistance development by animals and harmful impacts on human intestinal flora [50]. As the ingestion of xenobiotic substances can have negative impacts, it is prudent for food guidelines to clarify these concepts to their population [51].

Considering that the Healthy and Sustainable Diet Guidelines were effectively launched by the FAO in 2019, documents completed before this year were predictably more likely to be discordant in their recommendations. Nevertheless, the concept of “healthy diets” was officially endorsed by this organization in 2010 and has been the subject of academic and policy discussions since 2011 due to its emergence in the global health and climate crisis scenario [52,53]. Although its importance has been better addressed in recent years, it is not a new debate; the impact of the agri-food system on health and the environment has been intertwined since 1978 [53]. Since more than half of the food guidelines found in this study were released after 2010, there is a need to align the population’s guidance with current sustainability and health demands. Additionally, it is important to consider displaying this information in a way that is meaningful for caregivers of infants, taking into account that parents’ greatest concern must be the health and future of their children.

Nevertheless, even with less expressive scores in the Agree II and Healthy and Sustainable Diet Aspects evaluations, the food guides presented distinct and unique recommendations according to their populations and cultures. This characteristic is fundamental in the development of food-based dietary guidelines [11]. The literature points to the need to consider local food culture and various viewpoints on food in the development of food and nutritional education tools [2,3,4,54].

The Argentina guide proposed, in one recommendation, affection as an integral part of food and the benefits of a harmonious relationship with food. In the scientific literature, there is evidence that the relationship between life and food is a determining factor in eating disorder outcomes [55,56,57]. In this context, communal eating plays a protective role in promoting healthy eating habits by encouraging the consumption of homemade foods, fewer processed foods, and social cohesion [58].

Bolivia explicitly emphasized the importance of native ancestral foods in the dietary identity of its population. This recommendation aligns deeply with principles for reducing environmental impacts [48]. In local agri-food systems, short circuits of food production and consumption use fewer fossil fuels that generate greenhouse gases, thus reducing their environmental impact [48]. However, specific mentions of reducing meat consumption should complement the guidelines to achieve consonance with current sustainability requirements [53,54].

The Brazilian Population Food Guide stands out for its strong social participation in the guide’s elaboration process, which has been recognized as an important characteristic in other studies that oppose one of the major obstacles to the implementation of food guidelines [36,37,57]. Additionally, the Brazilian Guide showed the best evaluation performance due to the extent of its content, allowing for the coverage of more topics such as environmental impacts and social participation, resulting in a higher overall score. Throughout the document, there are directions for intersectoral public policies. Similarly, this was observed in the Colombia Guide, referring to health promotion and protection strategies, as well as the regulation of child advertising, topics addressed by the guides of Venezuela and Colombia.

References to legislation protecting children’s advertising were proposed as a strategy against aggressive marketing by the food industry, which has contributed to the devaluation of home cooking [36]. Advertising targeting children has a more damaging impact than targeting adults and induces the formation of nutritionally poor eating habits by associating media characteristics with ultra-processed foods [59].

One of the main differences found in the Chilean Food Guide for children under 2 years old was guidance on respectful weaning, with detailed recommendations providing support for this period. Respectful weaning is a strategy for the gradual weaning of a baby from breastfeeding, guided by physiological needs, motor skill development, and behavioral aspects of the infant [60]. In this approach, the baby becomes the protagonist of weaning and food introduction, and the authors propose that this approach increases the chances of establishing a good relationship with food from early life [60,61].

The food guides of Chile and Colombia provide recommendations to encourage the participation of both caregivers in the commitment to feeding children under two years old, including exclusive breastfeeding. In the literature, authors contribute a contemporary reflection on the defeminization of cooking and family care, as these roles are traditionally imposed on women compulsorily [62,63]. As instructive material for shaping eating habits, food guidelines have the potential to influence family feeding practices and promote gender role equity.

It is also important to note that a large part/most of the FBDGs found come from South America, which has a pioneering history in caring for nutritional health and food security in Latin America and the Caribbean [37]. In the geographical context of Latin America and the Caribbean, challenges arising from coloniality still foster a scenario of striking social inequalities, translated into social determinants [13,64]. In complementary feeding, this impact can be understood through indicators of feeding and socioeconomic indices [13]. Studies and reports show a relationship between income disparity and the prevalence of minimum acceptable feeding and minimum acceptable diversity in the populations of these countries [13,65]. Understanding that food introduction guidelines must be supported by intersectoral health and income strategies is key, as food security and the right to food are inseparable from the right to income.

Furthermore, as limitations of this study, the difficulty of accessing food guide documents on online search platforms can be considered. The outdated nature of these documents on the FAO search site hindered collection. This has hindered research and could be an obstacle to the interested public in reading these FBDGs. Additionally, although the Agree II analysis tool applies to various guidelines, including the context of health promotion, its goals are directed at clinical guidelines [18]. Some items (domain three, items nine and ten) are better applied to clinical guidelines than to population guidelines. Although the Fleiss kappa indicated moderate reliability, a fourth reviewer accompanied the analyses to establish an agreement. In future studies, an adaptation of the tool for a more accurate evaluation is suggested.

Additionally, the formulation, implementation, and evaluation of FBDGs must consider the perceptions of interested parties to evolve these documents in practice. Aligning the development and updates of guidelines with current global health needs and the perspectives of the targeted population may optimize their effective implementation. We strongly encourage future research to address the perspectives of caregivers of infants regarding the applicability of FBDGs.

## 5. Conclusions

The most highly evaluated FBDGs in this review were those with more recent release dates and greater quantitative content. These managed to encompass subjects related to health, sustainability, and citizenship within their scope. According to Agree II, the more robust guidelines received higher appropriateness scores, as they had a theoretical and methodological basis that ensured the recommendations. Therefore, updating the dietary guidelines for children under two in Latin America and the Caribbean is necessary to align them with current global health and sustainability requirements.

The various FBDGs presented in this study had diverse recommendations suitable for their populations. Future formulations and updates of dietary guidelines by countries and partner organizations can use this research as support. Additionally, providing language and health literacy support for this population is necessary. More revisions are required for FBDGs targeting specific populations such as children under 2 years of age, schoolchildren, adolescents, elderly individuals, and pregnant women to elucidate the needs and different strategies used in the health education of these groups.

## Figures and Tables

**Figure 1 nutrients-16-01233-f001:**
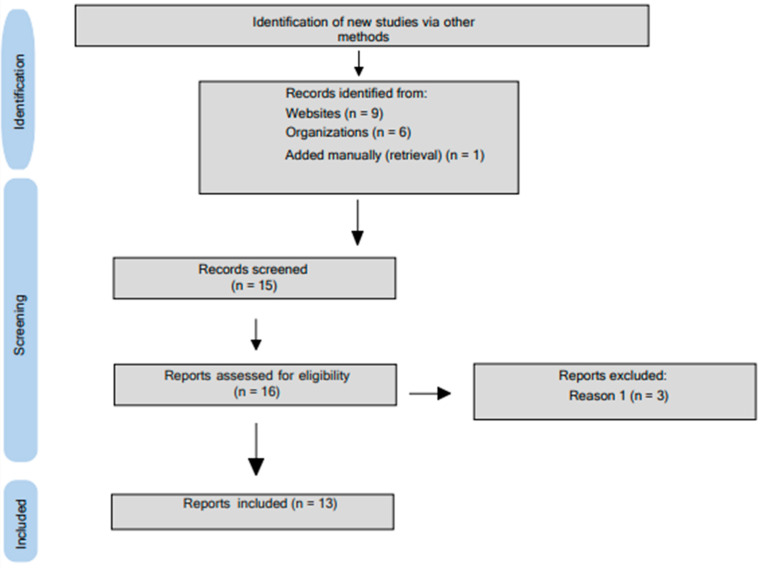
PRISMA fluxogram of document selection in gray literature. The PRISMA 2020 statement: an updated guideline for reporting systematic reviews. BMJ. Adapted.

**Table 1 nutrients-16-01233-t001:** Food-based dietary guidelines for infants, separated by country/region, launch data, number of key recommendations, pages, and abridged version.

Region/Country	Food-Based Dietary Guideline	Launch Year	Key Recommendations	Number of Pages	Abridged Version
Caribbean					
Dominican Repulic [22]	Lineamientos técnicos Guía de Alimentación Complementaria	2015	11	22	NO
Latin America					
Guatemala [21]	Guía Alimentaria para la población Guatemalteca menor de dos años	2003	17	94	NO
Panama [26]	Guías alimentarias para los menores de 2 años de Panamá.	2018	8	50	NO
Argentina [27]	Guías alimentarias para la población infantil: Orientaciones para padres y cuidadores	2009	9	50	NO
Bolivia [28]	Guía alimentaria para la mujer durante el período de embarazo y lactancia	2014	10	72	NO
Brazil [23]	Guia alimentar para crianças brasileiras menores de 2 anos	2019	12	265	YES
Chile [29]	Guía de alimentación de la niña y niño menor de 2 años y guía de alimentación hasta la adolescencia	2016	10	59	NO
Colombia [30]	Guías alimentarias basadas en alimentos para mujeres gestantes, madres en período de lactancia y niños y niñas menores de 2 años de Colombia	2020	17	107	YES
Mexico [31]	Guía Alimentación de la Familia	2012	2	94	NO
Paraguay [32]	Guías alimentarias del Paraguay para niñas y niños menores de 2 años	2015	8	29	YES
Peru [25]	Guías Alimentarias para niños y niñas menores de 2 años de edad	2020	19	44	NO
Uruguay [33]	Guía de alimentación complementaria para niños de entre 6 y 24 meses	2016	12	88	NO
Venezuela [34]	Alimentación en el Nivel de Educación Inicial	2011	7	101	NO

**Table 2 nutrients-16-01233-t002:** Percent adequacy scores for each domain of the Agree II assessment, organized by country, representing the FBDGs for children under two years old in the LAC.

FBDG (Country)	Domain (% Adequacy)						
	1. Scope and Purpose	2. Stakeholder Involvement	3. Rigor of Development	4. Clarity of Presentation	5. Applicability	6. Editorial Independence	7. Overall Assessment
Argentina	85%	65%	38%	93%	69%	56%	C 5
Bolivia	78%	48%	31%	65%	32%	56%	C 4
Brazil	100%	98%	76%	100%	82%	67%	C 7
Chile	91%	56%	42%	78%	53%	28%	C 5
Colombia	100%	87%	92%	80%	72%	56%	C 6
Guatemala	69%	52%	27%	54%	51%	50%	C 4
Mexico	59%	56%	25%	50%	65%	50%	C 4
Panama	98%	69%	29%	72%	51%	56%	C 5
Paraguay	100%	81%	38%	94%	54%	47%	C 5
Peru	100%	56%	49%	100%	68%	56%	C 6
Dominican Republic	89%	48%	31%	74%	46%	47%	C 4
Uruguay	98%	85%	49%	93%	76%	53%	C 6
Venezuela	61%	48%	35%	48%	56%	50%	C 4

**Table 3 nutrients-16-01233-t003:** Evaluation of FBDG key recommendations based on Sustainable Healthy Diets Guiding Principles (FAO), distributed by country.

Sustainable Healthy Diets Guiding Principles	Country/Guidelines												
Regarding the Health Aspect	ARG	BOL	BRA	CHL	COL	GUA	MEX	PAN	PAR	PER	DOM	URU	VEN
Orient exclusive breastfeeding until six months of age, combined with appropriate complementary feeding.													
Provide guidance on a variety of unprocessed or minimally processed foods, restricting highly processed food.													
Consumption of whole grains, legumes, nuts, and an abundance and variety of fruits and vegetables.													
Clarify moderate ingestion of eggs and red meat.													
Recommend drinking safe and clean water.													
Adequate energy and nutrients for growth and development and the needs of a healthy active lifestyle.													
Recommendations to reduce the risk of diet-related noncommunicable chronicle diseases.													
Orient appropriate hygiene to contain minimal/no levels of pathogens, toxins, and other agents of foodborne disease.													
Regarding Environment Impact													
Alert to the consequences of chemical pollution and how to help prevent them.													
Encourage the consumption of seasonal foods to preserve biodiversity resources.													
Offer guidance on the minimum use of antibiotics and hormones in food production.													
Encourage the minimal use of plastics and derivates in food packaging.													
Approach food loss with strategies to prevent food waste.													
Regarding Sociocultural Aspects													
Respect local culture, cuisine, practices, knowledge, and consumption patterns.													
Approach the accessibility of foods.													
Highlight the equal involvement of genders in home care to avoid gender-related impact.													

The legend of the colors: green if the guide provided an explanation and orientations about the item; yellow if there was only an explanation or orientation, and red if there was not found any explanation or orientation about the item.

## Data Availability

Data are contained within the article and Appendix A.

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
