# Peer review of "Food-Based Dietary Guidelines for Infants in Latin America and the Caribbean: A Systematic Review"

_nutrients, 2024, doi:10.3390/nu16081233_

Round 1

Reviewer 1 Report

Comments and Suggestions for Authors

The authors report a review of publicly available FBDG aimed at children (<2y) in Latin America and the Caribbean. The results show that more recent, and more detailed advice was more in line with FAO guide principles.

The main weakness of the report is that the guidelines are being evaluated by the authors who have extensive understanding of this subject, rather than the people that they are intended for – parents (who likely have a much lower level of initial understanding etc.). Perhaps this is why longer guidelines scored higher, although I can image that increasing length would be a barrier to parental understanding (and perhaps also engagement) of the documents. The authors specifically focused on guidelines intended for parents, not for professionals.

ABSTRACT

-          Please explain FAO at i’s initial use.

INTRODUCTION

-          The first paragraph really speaks to my previous concern that parental understanding of the document was not being assessed.

-          The introduction focuses on the importance of nutrition in early life. Most readers of the journal will be well aware of it. Perhaps the focus should be on the potential utility of FBDG to improve early childhood nutrition, and the barriers to achieving this.

MATERIAL AND METHODS

-          Line 76, is the term “gray literature” widely understood?

-          Line 96, “first semester”?

-          The review methodology seems to be evaluating the accuracy of the guidelines, rather than their utility.

-          Line 155, “near perfect agreement” seems better than “perfect reproducibility” as perfect agreement is a value of 1 (po – pe)/ (1 – pe) if observed agreement (po) is 1?

-          I didn’t get how the number in Figure 1 added up. We start with 16, 9 were excluded, but 16 were still assessed for eligibility, of who 13 were included?

-          I am not sure how many readers will be familiar with AGREE II, so perhaps a few lines on how to interpret the overall assessment in Table 2 would be helpful, perhaps within the legend of the Table or within the methods.

-          I was not surprised (frame 1) with the much lower scores for “environmental impact” than for “health aspects”. In order to be understandable and brief I would expect the guidelines to focus on what is of greatest interest to parents (health of the child) and what is most likely to motivate changes in behavior. Perhaps expecting them to address environmental impact in a document aimed to be readily understandable to parents is setting too high a goal.

DISCUSSION

-          I found the low percentage of countries that had parent facing nutritional guidance for children to be the most interesting result, and the authors are correct to start with this finding

-          Line 241. Perhaps these three countries understand that there long versions of guidelines are unlikely to engage parents, hence the shorted/ graphic version.

OTHER

-          There are too many references

Reviewer 2 Report

Comments and Suggestions for Authors

this review covering population of children under two years old in Latin America and the Caribbean (LAC), by using The Food-Based Dietary Guidelines (FBDGs) to 2023 several Guidelines aimed at caregivers of children, sourced from government websites  in LAC countries and the FAO portal. Authors used qualitative analysis, the Agree II guidelines assessment tool and the Food and Agriculture Organization (FAO) guide principles for developing healthy and sustainable diets were used. Results showed that more recently released and revised FBDGs with a higher number of pages obtained better scores in both assessments. The review is sound and necessary for latin America, including tables with Score of adequacies in percentage for each domain of Agree II, Evaluation of FBDG key recommendations based on Sustainable Healthy Diets Guide, check accurately english language

Comments on the Quality of English Language

check accurately english language

Round 2

Reviewer 1 Report

Comments and Suggestions for Authors

My main concerns remain, but as the authors comment are un-modifiable given the design used. The manuscript seems little changed from its prior version.

Other points:

Line 78: I think a date would be more understandable that “first semester”

Line 80: “the official government portals 79 of the countries (Supplementary Material 2) whose de guidelines were not included in the 80 databases were verified” is not correct grammatically.

Line 91: “As inclusion criteria were considered the official docu- 91 ments from departments of health, food, social and similar, of the 32 countries that form 92 the LAC, which the title specifically referred to “Food Guide for children under two years 93 old” or “Food Guide for pregnant Women and babies” or “Food Guide for the family” is not correct grammatically.

Figure 1: I still don’t think this makes sense. It seems to imply 1 report was sought for retrieval of which 16 were assessed for eligibility.

Comments on the Quality of English Language

There are minor grammatical errors, and some of the paper reads poorly, but the errors don't prevent understanding.
